# Synthesis of Dihydropyrimidines: Isosteres of Nifedipine and Evaluation of Their Calcium Channel Blocking Efficiency

**DOI:** 10.3390/molecules28020784

**Published:** 2023-01-12

**Authors:** Yasser M. Zohny, Samir M. Awad, Maha A. Rabie, Omar A. Al-Saidan

**Affiliations:** 1Pharmaceutical Sciences Department, College of Pharmacy, Shaqra University, Dawadmi P.O. Box 11961, Saudi Arabia; 2Pharmaceutical Organic Chemistry Department, Faculty of Pharmacy, Helwan University, Ein-Helwan, Cairo 11795, Egypt; 3Pharmacy Practice Department, College of Pharmacy, Shaqra University, Dawadmi P.O. Box 11961, Saudi Arabia; 4Pharmacology and Toxicology Department, School of Pharmacy, Cairo University, Cairo 11795, Egypt; 5Pharmaceutics Department, College of Pharmacy, Al-Jouf University, Sakaka P.O. Box 74666, Saudi Arabia

**Keywords:** antihypertensive agents, calcium channel blockers, dihydropyrimidins, cardiovascular diseases

## Abstract

Hypertension and cardiovascular diseases related to it remain the leading medical challenges globally. Several drugs have been synthesized and commercialized to manage hypertension. Some of these drugs have a dihydropyrimidine skeleton structure, act as efficient calcium channel blockers, and affect the calcium ions’ intake in vascular smooth muscle, hence managing hypertension. The synthesis of such moieties is crucial, and documenting their structure–activity relationship, their evolved and advanced synthetic procedures, and future opportunities in this area is currently a priority. Tremendous efforts have been made after the discovery of the Biginelli condensation reaction in the synthesis of dihydropyrimidines. From the specific selection of Biginelli adducts to the variation in the formed intermediates to achieve target compounds containing heterocylic rings, aldehydes, a variety of ketones, halogens, and many other desired functionalities, extensive studies have been carried out. Several substitutions at the C3, C4, and C5 positions of dihydropyrimidines have been explored, aiming to produce feasible derivatives with acceptable yields as well as antihypertensive activity. The current review aims to cover this requirement in detail.

## 1. Introduction

Currently, the whole world is still recovering from the COVID-19 (coronavirus disease, 2019) pandemic, which has led to millions of deaths. Hypertension has been recognized as one of the major and most common risk factor leading to adverse and severe outcomes in COVID-19 patients [1]. However, years prior to the outbreak of COVID-19, in the year 2003, “The Global Burden of Disease Study” from the World Health Organization (WHO) identified hypertension as a leading risk factor worldwide for morbidity as well as mortality [2]. Several cardiovascular diseases (CVDs) and even diabetes have been associated with hypertension, and serious concerns have been raised in the research community regarding the treatment and management of hypertension in human beings. Amongst the drugs developed for pharmacological applications, the contribution of heterocycles is tremendous. The base pairs of RNA and DNA are composed of heterocyclic structures, such as pyrimidine and purine [3]. Pyrimidines have established therapeutic properties in various cases, such as anti-inflammatory, antimicrobial, anticancer, and anti-analgesic effects.

Remarkably, since the synthesis of dihydropyrimidines (DHPMs) by the Italian chemist, Pietro Biginelli, in the year 1893, the determination of passionate researchers has led to modifications of the Biginelli condensation reaction, leading to the synthesis of DHPMs which can be used to achieve feasible yields and hassle-free experimental conditions. The efforts made to increase the diversity in the pharmacological activities of the derivatives of DHPMs containing pyrimidine scaffold have been tremendous [4]. However, after identifying DHPM scaffolds with antihypertensive activity as calcium channel antagonists, an abundance of literature has been produced. Plentiful research and review articles concentrating on calcium channel blockers [5,6,7,8,9], antihypertensive drugs [10,11,12], and the synthesis of DHPMs derivatives [13,14,15,16] with different uses are available from various platforms. The Web of Science (WOS) database was searched on 12 October 2022 with the keywords dihydropyrimidines or calcium channel blockers from the year 1980 to date, indicating a total of 51,647 publications on this topic. Figure 1 summarizes the number of publications produced per year as well as publications released in various fields of research. The current review, however, is focused on the same data due to the lack of review publications describing the synthesis of DHPMs in recent years. The present review article mainly considers the synthesis of DHPMs which are isosteres of nifedipine, providing an introduction of the different types of calcium channels and their modulators.

## 2. Calcium Channels

The calcium channels play a vital role in the normal functioning of the heart. Variations in their functioning disrupt the plateau phase of cardiac operations and may trigger irregular conduction cycles, resulting in cardiac dysfunction [17,18]. The primary purpose of the calcium channel is to carry calcium ions into cells [19,20]. The influx of calcium ions from the extracellular space to cells increases the positive potential of cell membranes, and the calcium ions released from the sarcoplasmic reticulum indicate the contraction of cardiac myocytes [21]. This process is widely known as “calcium-induced calcium release” (CICR) [22]. In anomalous cases, calcium channel modulators serve as significant pharmacological objectives, for which the study of the structural and functional features of the cardiac tissues is crucial. Depending on the pharmacological sensitivity and activation rate, so far, six types of calcium channels have been observed. These include T, L, N, P, Q, and R types [23], which are named according to their responses to the voltage applied. The most significant calcium channel subtypes are L-type and T-type, which play key roles in regulating heart functions. Both L-type and T-type are voltage-gated. L-type channels are long-lasting and their activation requires resilient depolarization, whereas T-type channels are transient [23]. A comprehensive understanding of the structural properties of these channels opens a wider pathway through which to identify their importance in cardiovascular pathologies.

L-type channels possess a long opening with a greater number of voltage-gated channels. In the course of membrane depolarization, the calcium influx occurs and offers an “excitation-contraction coupling”, in which L-type calcium channels are the crucial contributors [24,25]. These channels are expressed at the developing and adult stages of the heart. These channels are made of one pore-forming α sub-unit and another three β, δ, and γ accessory subunits. All these four subunits together will contribute to managing the current kinetics, membrane trafficking, and gating features [26]. In an early developmental stage, dilated left ventricles and restricted out-flux tracts are the result of the blocking of L-type channels.

In contrast to L-type channels, T-type channels are transient and feature low-voltage depolarization. With the development of the stages, the functionality of T-type channels decreases [27]. These are expressed in atrial cells rather than ventricular myocytes in the hearts of adults [28]. Similar to the L-type, T-type channels are composed of four homologous subunits containing S1 to S6 transmembrane helices [29]. Contributing to the pacemaker activity, their dysfunction would lead to bradycardia [30]. Some pathological complications are caused when T-type channels are present in atrial as well as ventricular myocytes [27]. When aberrant T-type channels are present, hypertrophied myocytes are seen, and blocking these channels increases cardiac fibrosis and impairs relaxation [30]. In order to rectify the issues associated with calcium channels, relying on the situation and demand, calcium channel modulators or blockers are designed and employed.

## 3. Calcium Channel Blockers

One in three persons faces hypertension, one of the primary heart-related disorders. Hypertension is a condition in which the heart pumps the blood with a force greater than the normal, pushing against the artery walls. There are seven types or classes of antihypertensive agents, and these are detailed in Figure 2. Each class has a different mechanism of action in hypertension reduction. The calcium channel blockers (CCBs) or calcium channel antagonists are such agents used to manage hypertensive conditions. These blockers avert the entry of calcium ions into the conducting cells and reduce the heart relaxation and contraction rate.

Under CCBs, there are two main categories: one is dihydropryridines and the other is non-dihydropyridines. Dihydropyridines mainly work in blood vessels and show lesser activity in the heart, whereas non-dihydropyridines act in the opposite way. As this review intends to focus on the dihydropyridimins as CCBs, the next sections will explain their mechanism of action and the procedures used for their synthesis.

## 4. Dihydropyrimidines as Calcium Channel Blockers

Three classes of CCBs can be identified: dihydropyridines, benzothiazepines, and phenylalkylamines. All of these vary in terms of their fundamental chemical skeleton as well as their relative selectivity for cardiac and vascular L-type channels. However, dihydropyridines are the most hassle-free and smooth muscle-selective CCBs. They are primarily used as hypertensive drugs, as they decrease the arterial pressure and systemic vascular resistance. This mode of action leads to reflux cardia stimulation and strongly enhances the demand of myocardial oxygen. The drug interaction at the cellular levels was studied extensively for the receptor sites of 1,4-dihydropyridines [31]. Nifedipine, nicardipine, nitredipine, amlodipine, and felodipine are some of the major CCBs which are used in clinical settings to treat cardiovascular disorders related to angina and hypertension [32]. Several SAR (structure–activity relationships) have been established for these molecules [33]. Aryl rings of these molecules substituted with different functionalities are positioned axially in the case of receptor-bound conformation. The synperiplanar orientation is preferred by C-4 aryl substituents. A cis-orientation is preferred by the ester group with respect to the C-5 and C-6 double bond. If the C-2 or C-6 substituents and the C-3 and C-5 esters are equivalent, then the molecule is said to have nonchiral *C_s_* symmetry. However, position C-4 is a chiral center with unsymmetrical substitutions. Moreover, unsymmetrical dihydropyridine enantiomers normally differ in their biological actions. Now and then, they can also show opposite actions similar to that of CCBs (calcium antagonists) v/s calcium agonists [34]. Although nifedipine is a highly successful drug, its short plasma half-life [35], which is due to the pyridine metabolic oxidation, is its major limitation [36]. Due to this, the frequent administration of the drug is crucial in achieving clinical efficacy. Substituted 2-chloro-1,4 dihydropryridine exhibited prolonged CCB activity compared to C-2 and C-6 methyl substituted 1,4 dihydropyridines [37].

Pyrimidines, which have similar structural skeletons of pyridines, are identified as the vital component of nucleic acid and play an integral role as biologically significant antitumor, antiviral, and cardiovascular agents. Unlike light-sensitive and aromatic 1,4-dihydropyridines, the N-3 substituted DHPMs are shown to possess promising stability [38]. Analogous to 1,4 dihydropyridines, DHPMs have been shown to have a boat conformation for the modulation of their activity. The ester carbonyl *cis* orientation and the up and down orientation of pseudoaxial aryl moieties substituted at the C-4 position of DHPM receptors suggest CCB as well as calcium channel agonistic activity, respectively [33].

The “functional interplay between DHPM receptor L-type calcium channels and ryanodine receptor calcium-release channel (R_y_R)” is highly important in SAR studies. The α_1S_ subunit of the DHPM receptor acts as a voltage sensor and signals the information directly to the RyR_1_ subtype, without the requirement of Ca^2+^ influx [39,40]. This kind of DCCR, a directly coupled calcium-release mechanism, is predicted to require physical contact between two associates. The coimmunoprecipitation of DHPM receptor containing α_1S_ and skeletal muscles solubilized by R_Y_R_1_ interaction is a strong basis for the above prediction [41]. Nevertheless, for this DCCR, the II-III cytoplasmic loop of α_1S_ is acknowledged as the most significant domain [42,43]. The opening of R_Y_R_2_ is induced by the opening of the α_1C_ subunit of the DHPM receptor due to the entry of extracellular Ca^2+^ ions upon depolarization [44]. This kind of calcium release induced by calcium ions again requires two partners at two positions, as the migration of Ca^2+^ towards R_y_R_2_ causes diffusion and buffering, which decreases the Ca^2+^ wave efficiency. It is assumed that α_1C_ is significantly engaged in the CICR coupling and, hence, not many other interactions between α_1C_ and R_Y_R_S_ have been explored. Meanwhile, there are few studies reporting the bidirectional strong coupling of DHPM receptors with R_Y_R in the cardiac muscle [45,46] as well as neurons [47]. The structural diversity of the DHPMs and their receptor subunits widely enriches their calcium blocking ability; therefore, their versatile synthesis has been investigated extensively.

## 5. Synthesis of Dihydropyrimidines

The investigation of dihydropyrimidines as CCBs was started in the mid-1980s with the examination of analogs **2** and **3**, which closely mimic nifedipine structure **1**, which is a scaffold of dihyropyridine (as shown in Figure 3) [48,49]. Although these molecules showed potential CCB action, their failure in in vivo antihypertensive activity was observed [50]. By then, there had been several attempts to structurally modify those analogues and to enhance their antihypertensive potency. Atwal et al. modified these molecules, obtained ester groups at N3 positions (**4**, **5**), and validated their antihypertensive activity [48]. However, the synthesized compounds were not orally active and upon further modifications, orally active compounds **6** and **7** were obtained.

Over several investigations, researchers have finally generalized the structure of DHPMs which showed potential against hypertension. The general structure is detailed in Figure 4. The structure–activity relationship studies reveal that *ortho* or *meta* substitutions in the phenyl ring contribute enormously to their in vitro potency. Aromatic substituents with electron-withdrawing groups and the ester alkyl group at C5 show the increasing potency in the order iso-propyl moieties > ethyl > methyl groups. Group “E” substituted at the nitrogen position is a firm requirement, with the *N*-alkyl carboxamido group in particular demonstrating better oral activity. Similarly, the order for X was S > O > N [51]. Though the generalized structure was investigated in the mid 1980s, the three-component reaction between aromatic aldehydes, urea, and acetoacetic esters was discovered by the Italian chemist P. Biginelli in 1893 to obtain 3,4-dihydropyrimidine-2(1H)-ones.

However, the reaction proceeds with low yields and requires a relatively long time (15–20 h) (Figure 1). A significant number of works have been devoted to the optimization of reaction conditions in order to increase the yields of target DHPMs. The effect of solvents and catalysts on the yields of the target products obtained in the Biginelli reaction have been studied recently. One approach used is the optimization of solvents (acetic acid, acetonitrile, THF, DMF, etc.) and the selection of appropriate catalyst systems (organic and inorganic acids, Lewis acids, ionic liquids, and many more). In order to accelerate the reaction, experiments have been performed with microwave irradiation, infrared irradiation, and ultrasonication, thereby reducing the reaction time to a few minutes and increasing the yield up to 98%. Huge numbers of aldehydes and protected aldehydes, some of them shown in Figure 5, were used in the cyclocondensation of Biginelli dihydropyrimidine synthesis.

In general, the reaction works best with aromatic aldehydes substituted in the *o-,-m-,* or *p-*positions with either electron-withdrawing or -donating groups [52]. Good yields are usually obtained with *m*- or *p*-substituted aromatic aldehydes [52]. For *o-*substituted benzaldehydes with bulky substituents, the yields can be significantly lower. Heterocyclic aldehydes derived from furan, thiophene, and pyridine rings generally provide acceptable yields of DHPM products [53]. Aliphatic aldehydes typically provide only moderate yields in the Biginelli reaction unless special reaction conditions are employed—i.e., Lewis acid catalysts/solvent-free methods—or when using the aldehydes in protected form. The C4 unsubstituted DHPM can be prepared in a similar manner by employing suitable formaldehyde synthons [54].

In addition, numerous CH-acidic carbonyl building blocks such as ethyl 3-oxopentanoate, 3-oxo-N-phenylbutanamide, N,N-diethyl-3-oxobutanamide, pentane-2,4-dione, 1-nitropropan-2-one, ethyl 4-bromo-3-oxobutanoate, 4-methyl-2H-1-benzopyran-2-one, cyclohexane-1,3-dione, ethyl 3-oxo-3-phenylpropanoate, 2-ethyl 3-oxobutanethioate, 2-chloroethyl 3-oxobutanoate, and methyl 3,3-dimethoxypropanoate can also be used in Biginelli reactions (Figure 6). Traditionally, simple alkyl acetoacetates are employed as CH-acidic carbonyl building blocks, but other types of 3-oxoalkanoic esters or thioesters can also be used successfully. For instance, the equivalent 6-chloromethyl-substituted DHPMs with 4-chloroacetate can act as important templates for additional synthetic transformations, as shown in Figure 2.

Benzoylacetic esters react analogously, but the yields are significantly lower and the overall condensation process is more sluggish (Figure 3). Primary, secondary, and tertiary acetoacetamides can be used in place of esters to produce pyrimidine-5-carboxamides, as predicted in Figure 4. In addition, ß-diketones serve as valuable substrates in Biginelli reactions. As shown in Zainab et al. 2012, the prepared compounds, for which the schemes are provided in Figure 5, showed promising anti-inflammatory activities [52]. Condensation can also be achieved via employing cyclic ß-diketones such as cyclohexane-1,3-dione and other cyclic ß-dicarbonyl compounds, as described in Figure 6. If a C6-unsubstituted DHPM derivatives need to be synthesized, the corresponding 3-oxopropanoic ester derivative in which the aldehyde function is masked as an acetal can be employed (Figure 7).

Nitroacetone is a useful building block for creating 5-nitro-substituted DHPM derivatives with good yields, in addition to ester-derived CH-acidic carbonyl compounds. The corresponding method is shown in Figure 8. Furthermore, the urea component in the Biginelli reaction faces the most restrictions in terms of the structural diversity allowed. Therefore, most of the published examples involve urea itself as a building block. However, simple monosubstituted alkyl urea generally reacts equally well in a regiospecific manner to provide good yields of N1-substituted DHPMs. Some similar units such as *N*-(prop-2-en-1-yl)urea, *N*-methylthiourea, *N*-phenylthiourea, methyl carbamate, (4-methoxyphenyl)methyl carbamimidothioate, and guanidine are provided in Figure 7.

Thiourea and substituted thioureas follow the same general rules. However, longer reaction times are required to achieve good conversions. In some instances, it is also possible to use protected urea, thioureas, or guanidines under weak basic conditions with the aldehyde and CH-acidic carbonyl component (or with a precondensed Knoevenagel-type enone) to yield the corresponding protected DHPMs, as shown in Figure 9. This latter method using precondensed enones has been frequently referred to as the ‘‘Atwal modification of the Biginelli reaction”. Twenty years ago, a huge number of DHPMs were synthesized and evaluated to determine their biological activities as antiviral; anti-tumor; antibacterial; anti-inflammatory; and, more recently, anti-hypertensive agents. In 2003, some Biginelli products were synthesized by Wageeh S. El-Hamouly et al. and screened to determine their antihypertensive activity (Figure 8). The findings obtained demonstrated that there was a strong anti-hypertensive activity in the series created using 4-chromonyl derivative. Among the synthesized compounds, the 4-chromonyl derivative showed a better anti-hypertensive potency than 4-pyrazolyl analogues and even than the standard drug nifedipine. With the aid of non-selective cat models, the group concluded that the dihydropyrimidines substituted with aryl functionalities were more potent than the dihydropyrimidones substituted with the same aryl functionalities.

Recently, some DHPMS were prepared as nifedipine isosteres and evaluated to determine their antihypertensive activity as calcium channel blockers. The current review is not exhaustive; therefore, recent attempts at synthesis with respectable levels of activity have been made. Mohamed Teleb et al. substituted N3 of DHPM and synthesized several derivatives for their CCB activity. Nifedipine was considered as the model structure and several modifications were made to improve the activity, safety profile, and pharmacokinetic activities of DHPMs [55,56,57]. Larger lipophilic alkyl and aryl groups positioned at the third or fifth in the form of esters show greater antihypertensivity [58].

Marwa and group synthesized 14 novel N3-substituted DHPMs and investigated their CCB activity [59]. Among them, the triazole-based DHPMs showed promising efficiency. The scheme used for the synthesis of all the compounds is provided in Figure 10. Ester compound A was hydrazinolysed to obtain compound B, which was used as the starting material for the synthesis. The compounds were further synthesized by heating the starting material with the corresponding 1,3-diaryl-1H-pyrazole-4-carbaldehydes, 1H-pyrrole-2-carbaldehyde, thiophen-2-carbaldehyde, heteroaromatic furfural, and 1-aryl1H-1,2,3-triazole-4-carbaldehydes in DMSO.

C2-substituted DHPMs are known to show optimal and desired pharmacokinetic activities. A 2(3H)-thioxo derivative of 1,4- 4 dihydropyrimidinethione was converted to 2-aroylmethylsulfanyl derivatives using phenacyl bromides in dry acetone [60]. Free amino derivatives, 3-hydroxypropylsulfanyl derivatives, were also synthesized correspondingly, as shown in Figure 11. 3-hydroxypropylsulfanyl derivative stirred in anhydrous pyridine and acetyl chloride offered an acetyl derivate and, upon reaction with 1.2 equivalents of ethyl chloroformate, gave a carbonate derivative. The same alcohol derivative was treated with benzenesulfonyl chloride in dry pyridine to give phenylsulfonate ester derivative and many more. The detailed synthesis scheme is provided in Figure 12. Several functionalities at C2 linked to the DHPM core could be synthesized by a variable spacer nature and size. These compounds act as CCBs via the whole-cell path clamp technique.

Introducing larger lipophilic alkyl as well as aryl DHPMs has interestingly been demonstrated to enhance safety and pharmacokinetic profiles [55,56,57]. These modifications are also able to harmonize the selectivity of tissues [61,62,63]. The same group synthesized several other target CCB moieties. Starting from three regular components of the Biginelli reaction, several target materials were synthesized via treatment with ethyl acrylate, KF/Al_2_O_3_, LiAlH_4_, THF, alcohols, and several other reagents, as shown in Table 1 [64]. The same research group has contributed enormously to the synthesis of DHPs and DHPMs [65,66]. They have attempted to synthesize CCBs, which could mimic ones which are of clinical use [4,21,22,23,24,25,27]. Several other modifications, such as triazole ligation, which introduce nitrile, carboxylic, and hydrazide groups and amino acid coupling reactions were investigated in order to obtain the compounds listed in Table 2 [67]. The DHPMs with hydrazine functionalities encaged in planar heterocylicpyrazole rings were found to be more efficient calcium channel blockers, while the nitro group derivatives presented a lesser efficiency [66].

## 6. Advanced Pathways

Along with the standard one-pot synthesis of DHPMs using the aforementioned three components and their variations under various reaction circumstances, more complex pathways were also investigated. Microwave-assisted synthesis is one such major success, and some microwave-assisted solvent-free reactions were also attempted. This section aims to highlight those advanced pathways.

### 6.1. Microwave-Assisted DHPMs Synthesis

The microwave-assisted synthesis of organic molecules and the combination of regular organic synthesis including high-throughput screening profiles are helpful in the quick synthesis of organic moieties with considerable levels of purity and yield [68,69,70,71,72,73,74]. A different set of aldehydes, urea, and β-ketoesters was created for the microwave-assisted synthesis of multifunctionalized DHPMs. The library generated by “automated sequential microwave-assisted Biginelli multicomponent condensation” was found to be efficient enough in terms of its diversity and combining speed [75,76]. Bimbisar Desai and group synthesized eleven compounds using microwave technology and Yb(OTf)_3_ as a Lewis acid [77]. Most of the substitutions were obtained at the C5 position of DHPM. The corresponding reaction conditions, obtained DHPMs, and their yields are listed in Table 3. The microwave-assisted deprotection of esters gave made it possible for the carboxylic acid bonds investigated to form multifunctionalized DHPMs. With the aid of polyphosphate ester (PPE) as the mediator of the reaction and coupled with microwave irradiation, various substituted DHPMs were synthesized with good yields, as described in Table 4 [78]. The same group has explored the replacement of C2-methylsulfonyl groups [79]. However, before these investigations, few publications had reported the displacement of the C2-sulphonyl group of pyrimidines with nucleophiles [80]. Additionally, there are some publications on the displacement of nucleophiles that are limited to primary and secondary amines [81]. There are plenty of DHPMs which are biologically active and which have been synthesized by microwave irradiation performed by C. Oliver Kappe and group [75,82,83,84]

### 6.2. Solvent-Free Synthesis of DHPMs

These simple Biginelli reactions have undergone numerous revisions. Microwave-assisted, solvent-free syntheses of DHPMs catalyzed by several efficient catalysts are trending at present. These are reactions carried out in dry media and can offer several benefits. The use of solvents in conventional Biginelli reactions makes it expensive or even toxic, and in some cases solvent removal is difficult, especially in the case of reactions based on aprotic solvents [85].

Ferric chloride hexahydrate has been identified as the effectual catalyst in C-C bond formation [86,87,88]. Maryam and group used this catalyst for revising the Biginelli reaction while keeping the three-component one-pot synthesis strategy. The group was successful in synthesizing microwave-assisted solvent-free DHPMs with considerably high yields [87]. In a typical DHPM synthesis, the corresponding three components of the Biginelli reaction were added to FeCl_3_·6H_2_O and the mixture was subjected to microwave irradiation at different time points. Water was added once the reaction mixture had reached room temperature. Ethyl acetate was used to extract the product and Na_2_SO_4_ was used for the drying procedure. Recrystallization was performed with ethylacetate to obtain pure products.

Researchers were also successful in solvent-free DHPM synthesis, without using any catalysts. Stefani and Gatti synthesized seven DHPMs using different aldehydes, as shown in Table 5; however, this was carried out without any solvents and catalysts [89]. Stefani presented a 30% yield of DHPM without adding any additives [89]. Mukut and group used CuCl_2_.2H_2_O as the catalyst, and the yield was increased from 20–50% [90] to 80–99%, while the reaction time reduced from 18–48 h to 1–1.5 min [91]. Xue Song et al. made an attempt to use different catalysts such as concentrated HCl, FeCl_3_.6H_2_O, SnCl_2_, ZnCl_2_, and CuCl_2_.2H_2_O [92]. Among all the catalysts, Lewis acids showed the most accelerated condensation (89–95% yields); however, the addition of one or two drops of concentrated HCl led to increased yields.

The use of dry acetic acid as a catalyst has produced enhanced efficiency due to its high polarity. The dry acid with a greater polarity helps in the adsorption of microwaves and the generation of heat energy. Yadav and group attempted to synthesize DHPMs using the same strategy and were successful in producing nearly 13 DHPM derivatives with a yield varying from 82 to 90% [93]. Likewise, NiCl_2_.H_2_O, LaCl_3_.7H_2_O [94], MgBr_2_ [95], iodine [96], and MgCl_2_.6H_2_O [97,98] are some other catalysts used for the DHPM synthesis.

### 6.3. Computational Investigations

Regardless of the increased number of patients with hypertension and associated CVDs, the efforts made in drug discovery in the present area are significant. Several SAR investigations strongly recommend the stereochemical and conformational requirements of the DHPMs to be used as calcium channel blockers [99,100]. A revisit of the literature helps us to determine the four significant parameters that play an important role in the biological activity of DHPMs. (1) The extent of the distortion of the aryl groups attached at the C-4 positions; (2) the cis and trans orientation of carbonyl groups positioned at C-3 and C-5, with respect to their subsequent double bonds; (3) the positioning of the aryl group and orthogonal arrangement; (4) the orientation of substitutions at the *ortho-* and *meta-* positions of the aryl group or any other equivalent position in the heterocycle ring with respect to C4-hydrogen [62,101]. Afshin Fassihi et al. have synthesized twenty symmetric as well as asymmetric DHPMs and analyzed their conformations using PM3 and DFT investigations [102]. Most of the structures were confirmed to have boat-like conformations. The PM3 method revealed the deviation of 54% of the molecules from their planes, but DFT calculations showed perfect flattened boat confirmations, similar to the nifedipine.

The quantity–structure–activity relationship (QSAR) and docking studies performed by Asghar Davood and group concluded that the symmetrical nature of the DHPM is not important, as only one imidazole nitrogen will interact with the receptor via hydrogen bonds [103]. The alterations in the imidazole rings and the addition of lipophilic groups to the *ortho-* or *meta-* positions of the heterocyclic ring enhance the CCB activity. Similar types of investigations were performed by Kappe and Walter; while ta similar conclusion was drawn, the structural conformations were analyzed by three different models—namely, ab initio, semi-empirical, and X-ray crystallographic models [104]. A greatly promising set of twenty-four training compounds was studied using pharmacophore models. Hydrogen donor–acceptor bonds and hydrophobic groups were the major components of that particular set, and these groups enhanced the calcium channel blocking [105]. Along with ab initio and semi-emperical models, molecular mechanic calculations were used for the comparative analysis of conformations of various derivatives of DHPM-based CCBs by Bahram and group [106]. Semi-empirical and molecular mechanic methods have shown the similar 3D structures and conformations of DHPMs, while the results provided by ab intio methods were very different. Most of the analyzed structures showed flattened boat-like conformations for DHPM rings, and very few molecules showed less deviated planarity. Studies did, however, confirm that the stability of molecules decreased when protons were added.

The in silico identification of selective and non-selective T-type calcium channel blockers was mapped via a ligand-pharmacophore approach for the first time in 2004 by Manikumar and group via the CTALYST software [107]. Authors have reported two important strategies and considered 10 test compounds for the validation of the results. In Strategy I, the hypothesis was generated by assuming that “all compounds are equally important and all contain important features”. Among the generated hypotheses, the mapping related to all the features of the active molecules to the maximum extent was considered as the best hypothesis. In strategy II, the hypothesis generation involved the bias between the most active compounds and the ones which presented poor fit values. The best fit hypothesis is as provided in Figure 9.

Similarly, six different chemical properties are considered in this study: aromatic ring, hydrophobic aliphatic, two hydrophobic aromatics, hydrogen bond donor and acceptor, and finally positive ionizable characteristics [108]. The molecules which are selective CCBs showed all six features. Meanwhile, the non-selective molecules showed five features but not the aromatic ring property, as shown in Figure 10.

## 7. Other Therapeutic Accomplishments of Dihydroyrimidines

While DHPMs have been explored extensively as calcium channel modulators, humankind is fortunate to have a scaffold such as DHPMs, as it possesses other diverse pharmacological applications. Along with calcium channel modulators, the appropriate decoration of various functional groups furthers its applications for the blockage of K^+^ ions activated by Ca^2+^ ions in human erythrocytes, for which the best examples are nifedipine and its analogues [109]. (+) Niguldipine has been proved to activate Ca^2+^-dependent, large conductance K channels (BK_Ca_) in “human mesenteric vascular smooth muscles”. Some of these DHPMs have also been identified to block bladder KATP channels and used in the case of urinary issues [110]. Calcium channel antagonists are identified as anticonvulsant agents and could encourage seizures with respect to their dosage [111,112].

The contribution of the lipophilic derivatives of DHPMs to the selective delivery of drugs to the brain has been revealed [111,113]. The delivery of valproic and valeric acids to the brain was investigated by Yiu and Knaus [111]. An equilibrium between DHPM and the pyridinium salt-type redox system was used by El-Sherbeny et al. to deliver the cytotoxic drugs to the brain [114]. The antitubercular activity of DHPMs containing lipophilic dicarbamoyl groups against *Mycobacterium tuberculosis H37Rv* was investigated [115,116]. Neovascularization induced by TNF-α was found to be inhibited by nifidepine and its analogues [117]. The antioxidant features, GABA receptor inhibition, and analgesic properties [118]; hepatoprotective [119,120]; antifungal [121,122]; antiplatelet [123,124]; and bronchodilatory activities [125] are other features of DHPMs depending on their decorated functionalities and substituents.

## 8. Conclusions and Future Perspective

Several modifications and revisions of Biginelli reactions have been developed and evolved by enthusiastic researchers and an enormous amount of literature is available. Yet, HMPMs continue to be a promising field, as the pyrimidine scaffold offers diverse pharmacological applications for the DHPM derivatives. The presence of several other heterocycles, such as imidazole, thiazole, pyrazole, benzothiazole, phenothiazine, indole, and many more, along with pyrimidine scaffold, have increased the anti-hypertensive property of the DHPMs. The investigations also suggest that functional groups such as –NO_2_, -F, -NHCO, -Cl, -Br, -CF_3_, -COOCH_3_, -OCH_3_, -OC_2_H_5_, and several groups are found to enhance the biological features of DHPM derivatives. Combinatorial syntheses, efforts towards the optimization of yields, accomplishing green catalysts, and the creation of synthetic methods to reduce the danger of emerging hazardous chemicals are still topics worthy of investigation. Additionally, the recognition of suitable substitutions leading to lower toxicity and the best pharmacological activity would be a first step towards overcoming the toxicity issues. The deprotection of esters of the Biginelli adducts assisted by micro-waves led to the efficient synthesis of the DHPM derivatives. In the case of such solvent-free synthesis, the addition of small amount of acids resulted in a greater yield. The pharmacophore models suggest that the hydrogen donor and acceptor substituents and hydrophobic substituents present good calcium channel blocking abilities. The employment of green chemistry and catalysts; avoiding solvents; and the synthesis of new skeletons containing bi- and tricyclic structural parameters such as “benzo [4,5] imidazo [1,2-*a*]-pyrimidine, 4H-pyrimido [2,1-*b*]benzothiazole, and 5,8-dihydro-[1,2,4]triazolo [4,3-*a*]pyrimidine” could be explored. In order to establish the promising activity of synthesized drugs, QSAR techniques must be improved in order to achieve more accurate binding studies.

This literature review of the use of Biginelli reactions for the synthesis of DHPM derivatives suggests that enantioselective reaction adducts and conditions result in efficient anti-hypertensive products. The synthesis of optically active DHPMs is popular, in addition to the above-mentioned features and techniques; however, this remains unexplored in the present review.

## Data Availability

Not applicable.

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
