# Peer review of "Synthesis of Dihydropyrimidines: Isosteres of Nifedipine and Evaluation of Their Calcium Channel Blocking Efficiency"

_molecules, 2023, doi:10.3390/molecules28020784_

Round 1

Reviewer 1 Report

This review article is a simplistic literature review, but not an insightful one. It is citing publications and reporting their discoveries without analysis or perspectives. There is no research or clinical viewpoints either in the narrative or the conclusions. The figure seem to be direct adaptions from reviews and figure legends are either vaguely descriptive or not descriptive at all. 

• What is the main question addressed by the research?

The article aims to focus on synthesis of DHPMs and their use as of calcium channel modulator.

• Do you consider the topic original or relevant in the field? Does it address a specific gap in the field?

While the topic is relevant, I do not believe it is presented in a way that is largely helpful to the field. This article is simply quoting material from other articles without a critical analysis.

• What does it add to the subject area compared with other published material?

Although this article has a lot of material put in, it doesn’t provide insights into what is already known. For example figure 6 lists "Figure 6. Some of the aldehydes and protected aldehydes used in the Biginelli reaction” What is the relevance of simply listing all these structured tried in a previously published paper?

• What specific improvements should the authors consider regarding the methodology? What further controls should be considered?

The authors need to justify why they are simply stating all these structures and papers. Simply using data published in other papers and describing them in text wouldn’t be sufficient for a review in that field.

• Are the conclusions consistent with the evidence and arguments presented and do they address the main question posed?

The conclusion is very small compared to the data presented all across the article and doesn’t do any justice to the field, the role of DHPMs as calcium channel modulators!

• Are the references appropriate?

It is acceptable

• Please include any additional comments on the tables and figures.

The table and figures do not have appropriate legends/descriptions. For example, "Figure 17. Some Biginelli products synthesized by Wageeh S.El-Hamouly et al. “ That is not descriptive at all. A figure legend needs to explain what is in the figure at a minimum.

Author Response

Dear Mr. Djordje Toljic

Authors thank the reviewers and editors for their useful comments and suggestions, which helped the authors to improve the manuscript. All the comments are addressed and tracked in the revised manuscript and authors hope that, the revisions are satisfactory and now could be considered for the publications.

Manuscript ID: molecules-2001744

Type of manuscript: Review

Title: Synthesis and Anti-hypertensive Evaluation of Dihydropyrimidines as
Calcium Channel Blockers: Isosteres of Nifedipine

Reviewer 1

This review article is a simplistic literature review, but not an insightful one. It is citing publications and reporting their discoveries without analysis or perspectives. There is no research or clinical viewpoints either in the narrative or the conclusions. The figure seem to be direct adaptions from reviews and figure legends are either vaguely descriptive or not descriptive at all. 

Response: The analysis for the cited literature is added then and there and in the conclusions as well. The future perspectives are also included, the structures in the figures are now replaced by the ones drawn from Chemsketch and the figure legends are now more descriptive in the revised manuscript. We hope the suggestions are addressed and authors thank the reviewer for the useful suggestions.

  • What is the main question addressed by the research?

The article aims to focus on synthesis of DHPMs and their use as of calcium channel modulator.

  • Do you consider the topic original or relevant in the field? Does it address a specific gap in the field?

While the topic is relevant, I do not believe it is presented in a way that is largely helpful to the field. This article is simply quoting material from other articles without a critical analysis.

Response: Thank you for the concern. Authors hope that, the critical analysis done and tracked in the revised manuscript are satisfactory.

  • What does it add to the subject area compared with other published material?

Although this article has a lot of material put in, it doesn’t provide insights into what is already known. For example figure 6 lists "Figure 6. Some of the aldehydes and protected aldehydes used in the Biginelli reaction” What is the relevance of simply listing all these structured tried in a previously published paper?

Response: As the present review article concentrates on the synthesis of dihydropyrimidines, which act as calcium channel blockers, authors feel that, listing out the basic and versatile reactants used in the previously published articles would help the readers in understanding the selection of the reactant molecules.

  • What specific improvements should the authors consider regarding the methodology? What further controls should be considered?

The authors need to justify why they are simply stating all these structures and papers. Simply using data published in other papers and describing them in text wouldn’t be sufficient for a review in that field.

 Response: The required critical analysis is done in the revised manuscript and the changes are tracked for the easy identification. Authors hope that, the revisions are satisfactory and now could be recommended for the considerations.

  • Are the conclusions consistent with the evidence and arguments presented and do they address the main question posed?

The conclusion is very small compared to the data presented all across the article and doesn’t do any justice to the field, the role of DHPMs as calcium channel modulators!

Response: The conclusions are now extended and more insights to the interpreted data as well as future perspectives are added in the revised manuscript.

  • Are the references appropriate?

It is acceptable

  • Please include any additional comments on the tables and figures.

The table and figures do not have appropriate legends/descriptions. For example, "Figure 17. Some Biginelli products synthesized by Wageeh S.El-Hamouly et al. “ That is not descriptive at all. A figure legend needs to explain what is in the figure at a minimum.

Response: Thank you very much the comment. The figure legends are now made more descriptive in the revised manuscript.

Reviewer 2 Report

Manuscript Title: Synthesis and Anti-hypertensive Evaluation of Dihydropyrimidines as Calcium Channel Blockers: Isosteres of Nifedipine

Manuscript ID: molecules-2001744

Article Type: Review

Comments:

The manuscript “Synthesis and Anti-hypertensive Evaluation of Dihydropyrimidines as Calcium Channel Blockers: Isosteres of Nifedipine” by Yasser M. Zohny et al drafted a review for the synthesis of several Dihydropyrimidines scaffolds for the anti-hypertensive evaluation which are isosteres to the Nifedipine.

This review was poorly written with many grammatical errors. The authors need to keep proper attention while writing and the errors were highlighted in an attached file. The authors should be more cautious and careful while communicating the review. It is so important that authors need to pay more attention and read through the related manuscripts and try to understand the basics and fundamentals of science and justify it.

Please read some of the recent articles published in ‘Molecules’ related to your work. After careful reading, I am considering this manuscript needs a minor revision. Please re-submit the manuscript with the improved and revised version.

Comments:

1This review should be corrected by an English expert. The chemistry figures and tables are not properly drawn.

2.      The title needs to be changed and improperly written. The sentence framing is also not good. The “Synthesis and Anti-hypertensive Evaluation of Dihydropyrimidines as Calcium Channel Blockers: Isosteres of Nifedipine”, I didn’t understand the connectivity between the two sentences. Please revise it.

3.      The abstract needs to be added much more context and to be revised with a detailed explanation.

4.      The text for each figure within this review needs to be given more explanation.

5.    The chem draw files should be upgraded and drawn with the standard format. All figures need to be drawn in a standard format and pasted into an original format.

Author Response

Dear Mr. Djordje Toljic

Authors thank the reviewers and editors for their useful comments and suggestions, which helped the authors to improve the manuscript. All the comments are addressed and tracked in the revised manuscript and authors hope that, the revisions are satisfactory and now could be considered for the publications.

Manuscript ID: molecules-2001744

Type of manuscript: Review

Title: Synthesis and Anti-hypertensive Evaluation of Dihydropyrimidines as
Calcium Channel Blockers: Isosteres of Nifedipine

Reviewer 2:

Manuscript ID: molecules-2001744

Article Type: Review

Comments:

The manuscript “Synthesis and Anti-hypertensive Evaluation of Dihydropyrimidines as Calcium Channel Blockers: Isosteres of Nifedipine” by Yasser M. Zohny et al drafted a review for the synthesis of several Dihydropyrimidines scaffolds for the anti-hypertensive evaluation which are isosteres to the Nifedipine.

This review was poorly written with many grammatical errors. The authors need to keep proper attention while writing and the errors were highlighted in an attached file. The authors should be more cautious and careful while communicating the review. It is so important that authors need to pay more attention and read through the related manuscripts and try to understand the basics and fundamentals of science and justify it.

Please read some of the recent articles published in ‘Molecules’ related to your work. After careful reading, I am considering this manuscript needs a minor revision. Please re-submit the manuscript with the improved and revised version.

Comments:

1This review should be corrected by an English expert. The chemistry figures and tables are not properly drawn.

Response: Thank you for the comment. Some of the unclear figures and tables are replaced with the good quality ones in the revised manuscript.

  1. The title needs to be changed and improperly written. The sentence framing is also not good. The “Synthesis and Anti-hypertensive Evaluation of Dihydropyrimidines as Calcium Channel Blockers: Isosteres of Nifedipine”, I didn’t understand the connectivity between the two sentences. Please revise it.

Response: The title has been now revised to “Synthesis of Dihydropyrimidines: Isosteres of Nifedipine and evaluation of their Calcium Channel Blocking efficiency”.  Authors hope that, the revised title is now more understandable.

  1. The abstract needs to be added much more context and to be revised with a detailed explanation.

Response: Thank you very much for the comment, which helped us to improve the abstract quality. More context is now added in the abstract and the new abstract is pasted below.

Hypertension and related cardiovascular diseases remain as the leading challenges globally. Several drugs are synthesized and commercialized to manage the hypertension. Some of the drugs are with the dihydropyrimidine skeleton structure act as efficient calcium channel blockers and accomplish the calcium ions intake to vascular smooth muscle and hence managing the hypertension. Synthesis of such moieties is crucial for this time and a document detailing their structure activity relationship, evolved and advanced synthetic procedures and future opportunities are the needs of the hour. Tremendous efforts have been identified after the discovery of Beginelli condensation reaction for the synthesis of dihydropyrimidines. From the specific selection of Beginelli adducts to the variation in the formed intermediates to achieve target compounds containing heterocylic rings, aldehydes, verity of ketones, halogens and many other desired functionalities have been studied extensively. Several substitutions at the C3, C4, C5 positions of dihyfropyrimidines to result in feasible derivatives with acceptable yield as well as anti-hypertensive activity are explored. Current review aims to cater the requirement in detail.

  1. The text for each figure within this review needs to be given more explanation.

Response: The legends of the figures are now more descriptive and tracked in the revised manuscript.

  1. The chem draw files should be upgraded and drawn with the standard format. All figures need to be drawn in a standard format and pasted into an original format.

Response: Thank you for the comment. Previously pasted chem draw structures were redrawn and pasted from the original format in the revised manuscript.

Thanking you

Round 2

Reviewer 1 Report

many of my comments have been addressed. paper can be accepted with English style revision.

Author Response

Dear Dr.

Thank you for your remarks/corrections. We followed your corrections and corrected the errors. All the corrections are taken into consideration in the attached file.

Best regards,
